# A Simultaneous Conversion and Extraction of Furfural from Pentose in Dilute Acid Hydrolysate of *Quercus mongolica* Using an Aqueous Biphasic System

Jong-Hwa Kim [1], Seong-Min Cho [1], June-Ho Choi [1], Hanseob Jeong [2], Soo Min Lee [2], Bonwook Koo [3,*] and In-Gyu Choi [4,5,*]

1    Department of Forest Sciences College of Agriculture and Life Sciences, Seoul National University, Seoul 08826, Korea; wmfty@snu.ac.kr (J.-H.K.); csmin93@snu.ac.kr (S.-M.C.); jhchoi1990@snu.ac.kr (J.-H.C.)
2    Wood Chemistry Division, Forest Products Department, National Institute of Forest Science, Seoul 02455, Korea; hsj17@korea.kr (H.J.); lesoomin@korea.kr (S.M.L.)
3    Green and Sustainable Materials R & D Department, Korea Institute of Industrial Technology, Cheonan-si 31056, Korea
4    Department of Agriculture, Forestry, and Bioresources, College of Agriculture and Life Sciences, Seoul National University, Seoul 08826, Korea
5    Research Institute of Agriculture and Life Sciences, College of Agriculture and Life Sciences, Seoul National University, Seoul 08826, Korea
*    Correspondence: bkoo@kitech.re.kr (B.K.); cingyu@snu.ac.kr (I.-G.C.)

**Abstract:** This study optimizes furfural production from pentose released in the liquid hydrolysate of hardwood using an aqueous biphasic system. Dilute acid pretreatment with 4% sulfuric acid was conducted to extract pentose from liquid *Quercus mongolica* hydrolysate. To produce furfural from xylose, a xylose standard solution with the same acid concentration of the liquid hydrolysate and extracting solvent (tetrahydrofuran) were applied to the aqueous biphasic system. A response surface methodology was adopted to optimize furfural production in the aqueous biphasic system. A maximum furfural yield of 72.39% was achieved at optimal conditions as per the RSM; a reaction temperature of 170 °C, reaction time of 120 min, and a xylose concentration of 10 g/L. Tetrahydrofuran, toluene, and dimethyl sulfoxide were evaluated to understand the effects of the solvent on furfural production. Tetrahydrofuran generated the highest furfural yield, while DMSO gave the lowest yield. A furfural yield of 68.20% from pentose was achieved in the liquid hydrolysate of *Quercus mongolica* under optimal conditions using tetrahydrofuran as the extracting solvent. The aqueous and tetrahydrofuran fractions were separated from the aqueous biphasic solvent by salting out using sodium chloride, and 94.63% of the furfural produced was drawn out through two extractions using tetrahydrofuran.

**Keywords:** aqueous biphasic system; dilute acid hydrolysate; furfural production; solvent extraction; response surface methodology

## 1. Introduction

Lignocellulosic biomass is considered as an alternative energy resource that can mitigate the climate change associated with the excessive use of fossil fuels [1]. Owing to its abundance and non-edibility, cellulose in particular, the key component of biomass, is a rich source of carbohydrates and has been applied to value-add to chemicals or biofuels [2]. Hemicellulose and lignin, the other main components of biomass, combine complex and dense forms of cellulose [3]. This physical barrier makes lignocellulosic biomass chemically and microbiologically resistant [4]. This recalcitrance of biomass requires a pretreatment process to ensure that lignocellulosic biomass is utilized properly and efficiently. The purpose of the pretreatment process is to cleave lignin and hemicellulose and obtain cellulose to improve accessibility for chemicals or enzymes [5]. Among several pretreatment methods, dilute acid

pretreatment is considered to be a leading pretreatment technology. This is because it is able to enhance the total sugar yield of the process by solubilizing and converting hemicellulose to pentose [6]. As the pretreatment of lignocellulosic biomass is highly demanding in terms of energy and additional processes compared to edible biomass, its economic feasibility is reduced [7]. As such, utilization of all the main components of biomass to generate valuable products is essential to make this resource more economically feasible.

Hemicellulose is a heteropolysaccharide composed of various monosaccharides such as xylose, arabinose, and mannose [8]. Traditionally, it is considered a by-product in the pulping industry, whereby most hemicellulose is dissolved into black liquor and used as a heating source [9]. However, pentose in hemicellulose may also be converted to valuable chemicals in the same manner as cellulose. This will subsequently improve the economic feasibility of the biorefinery industry through the utilization of lignocellulosic biomass.

Furfural is a building block chemical applied to various fields such as fuel, chemicals, polymers, and pharmaceuticals. Most furfural is produced by acid-catalyzed dehydration of pentose derived from hemicellulose [10]. It may also be produced from lignocellulosic biomass directly via acid catalyst treatment. Although this is a simple and mature process, it is characterized by several disadvantages including low furfural yield, generation of undesirable by-products, and difficulties in utilizing the remaining biomass, such as cellulose or lignin [11]. For this reason, many studies have proposed a two-step process in which hemicellulose is hydrolyzed into pentose or pentose-derived oligomers, and then the pentose or the oligomer is dehydrated into furfural in different reaction systems. This separated system offers several advantages including being able to produce a high furfural yield via the optimization of the reaction system for furfural production (catalyst, solvent, reaction condition, etc.) [12]. The separation of furfural from reaction media continues to be a challenge due to several complicated processes [13].

Furfural may be degraded and condensed in the presence of an acidic catalyst and water [14]. In an aqueous furfural production system that uses water as a solvent, the degradation and self-condensation of furfural limits high yields of furfural. This is despite the fact that water is a commonly used solvent for furfural production because it is inexpensive and eco-friendly. To solve this problem, furfural must be separated from the aqueous solvent system immediately after conversion from pentose.

An aqueous biphasic system may be used to separate furfural from water in the system. The aqueous phase contains water, and the acid catalyst converts pentose to furfural, while the organic phase is composed of the organic solvent that absorbs furfural converted into the organic phase. Dichloromethane (DCM) [15], methylisobuthylketone (MIBK) [16], and tetrahydrofuran (THF) [17] have been used as organic solvents for the extraction. These organic solvents are advantageous in terms of conducting a simultaneous process, including the conversion of pentose to furfural and the extraction of furfural. Furfural is produced by an acid catalyst in the aqueous phase and is extracted immediately by the organic phase [18]. Organic solvents may minimize furfural degradation, improving furfural yield [19].

This study aims to optimize furfural production from the dilute acid hydrolysate of *Quercus mongolica*. A xylose standard solution with the same acid concentration of dilute acid hydrolysate and extracting solvent (THF) were applied to the aqueous biphasic system to determine the optimal conditions for furfural production. A response surface methodology (RSM) in which the independent variables were reaction temperature, time, and xylose concentration was adopted to optimize the aqueous biphasic system. The organic solvent was also evaluated to select the solvent that prevented the degradation and self-condensation of furfural.

## 2. Materials and Methods

### 2.1. Materials

The xylose standard was purchased from Sigma-Aldrich Korea Co. (Yongin, Republic of Korea). *Quercus mongolica*, a xylan-rich hardwood was used as the raw material for

pentose and it was supplied by the National Institute of Forest Science (NIFoS, Seoul, Republic of Korea). The particle size of the raw material was reduced to less than 0.5 mm through grinding and sieving using a sawdust producer and air classifier mill, respectively. The moisture content was less than 5%, and the chemical composition was determined using the Laboratory Analytical Procedure of National Renewable Energy Laboratory (NREL, Golden, CO, USA) [20].

### 2.2. Dilute Acid Pretreatment for Pentose Production from Lignocellulosic Biomass

Dilute acid pretreatment of *Quercus mongolica* for pentose production was conducted following the methodology described in previous research [21]. Briefly, raw material was mixed with sulfuric acid solution (4%, $w/w$) in an Erlenmeyer flask; the solid to liquid ratio was 1 to 7. Then, the flask was placed in an autoclave (MLS-3020, Sanyo, Osaka, Japan) at 121 °C for 102.3 min; these are the optimal conditions for pentose production from *Quercus mongolica* as described in the previous research [21]. Following the dilute acid pretreatment, the flask was immediately cooled to room temperature in the ice chamber to stop the reaction. Then, the pentose-rich hydrolysate was separated from the solid residue using a Büchner funnel equipped with filter paper (No. 52, Hyundai Micro Co., Seoul, Korea). The chemical composition of the hydrolysate is shown in Table 1.

**Table 1.** Chemical composition of dilute acid hydrolysate of *Quercus mongolica*.

| Component | Concentration (g/L) |
| --- | --- |
| Sugars | |
| Glucose | $2.01 \pm 0.05$ |
| Xylose+mannose+galactose (XMG) | $20.02 \pm 0.46$ |
| Arabinose | $1.64 \pm 0.06$ |
| Sugar derivatives | |
| Acetic acid | $6.17 \pm 0.13$ |
| Formic acid | $0.03 \pm 0.00$ |
| Furfural | $0.32 \pm 0.04$ |
| 5-Hydroxymethylfurfural (5-HMF) | $0.01 \pm 0.00$ |

### 2.3. Response Surface Methodology for Optimization of Furfural Production from Xylose Standard

To maximize furfural production from the hydrolysate, the optimum conditions needed to be determined; optimization was conducted using a xylose standard solution. The xylose standard solution was adopted to investigate the relationship between reaction conditions and furfural yield, excluding the effect of impurities. Briefly, 10 mL of xylose standard solution containing a certain concentration of xylose was mixed with 20 mL of organic solvent in a Teflon-lined reactor. The sulfuric acid concentration of the xylose standard solution was adjusted to 4% ($w/w$); this was the same as the dilute acid hydrolysate used to obtain the optimum reaction conditions for the hydrolysate. The reactor was then sealed and soaked in an oil bath that had been pre-heated to a target temperature. The mixed hydrolysate was stirred during the reaction using a magnetic stirrer, and the temperature was maintained for a certain reaction time at the target temperature. After the reaction, the reactor was removed from the oil bath and immediately stored in the ice chamber to cool to room temperature and prevent undesirable reactions.

An RSM was adopted to optimize furfural production using a xylose standard solution. The analysis was conducted based on a $2^3$ factorial central design (CCD) using Design Expert 11.1.0.1 software (Stat-Ease, Inc., Minneapolis, MN, USA). Table 2 presents 17 sets of reaction conditions composed of six axial points and a duplication of the central point. The reaction temperature (X1, °C), reaction time (X2, min), and xylose concentration (X3, g/L) were designated as independent variables, while furfural yield (Y1, %) was the dependent variable.

**Table 2.** $2^3$ factorial experimental design varying on 3 factors and results of furfural yield.

| No. | Independent Variables | | | Dependent Variable |
|---|---|---|---|---|
| | Reaction Temperature $(X_1, {}^\circ C)$ | Reaction Time $(X_2, min)$ | Xylose Concentration $(X_3, g/L)$ | Furfural Yield $(Y_1, \%)$ |
| 1 | 140 | 60 | 10 | 6.96 |
| 2 | 180 | 60 | 10 | 69.87 |
| 3 | 140 | 180 | 10 | 39.47 |
| 4 | 180 | 180 | 10 | 62.48 |
| 5 | 140 | 60 | 30 | 6.67 |
| 6 | 180 | 60 | 30 | 66.83 |
| 7 | 140 | 180 | 30 | 46.59 |
| 8 | 180 | 180 | 30 | 59.23 |
| 9 | 126.36 | 120 | 20 | 4.69 |
| 10 | 193.64 | 120 | 20 | 61.33 |
| 11 | 160 | 19.09 | 20 | 0.71 |
| 12 | 160 | 220.91 | 20 | 68.12 |
| 13 | 160 | 120 | 3.18 | 67.99 |
| 14 | 160 | 120 | 36.82 | 67.77 |
| 15 | 160 | 120 | 20 | 67.75 |
| 16 | 160 | 120 | 20 | 67.75 |
| 17 | 160 | 120 | 20 | 68.02 |

The coded level of the CCD from each run was applied to real independent variables as follows:

Variable = value of central point/variation of coded level per one point

Reaction temperature (°C) = 160/20, reaction time (min) = 120/60, xylose concentration (g/L) = 20/10

Following the optimization of reaction conditions based on the RSM, the extraction solvent was altered from THF to dimethyl sulfoxide (DMSO) or toluene in the optimum conditions to analyze the effects of the extraction solvent.

### 2.4. Furfural Production from Pentose Derived from Quercus Mongolica

Simultaneous reactions occurred involving the release of pentose from xylose and the conversion of pentose to furfural in the same reactor without a separation process; this places it in a good stead for industrial application. The pentose derived from xylose was converted to furfural in the same reactor, used for pentose release. The dilute acid hydrolysate of *Quercus mongolica* was mixed with 4% sulfuric acid to adjust xylose concentration to optimal conditions in the Teflon-lined reactor; the total volume was adjusted to 10 mL. Then, 20 mL of THF (i.e., the organic solvent), was added into reactor for furfural extraction. The reaction temperature and time were set to the optimal values as per the results from the RSM analysis. After the reaction, the reactor was removed from the oil bath and immediately stored in the ice chamber to cool to room temperature and prevent undesirable reactions. The organic phase of THF, containing furfural, was separated from the aqueous phase through the addition of NaCl. This was done to investigate the separation efficiency of furfural from the final solution, including furfural and other products.

### 2.5. Analysis of Furfural and Other Products in Mixed Hydrolysate

The content of furfural and other products such as the remaining sugars was determined using high performance liquid chromatography (Ultimate-3000, Thermo Dionex, Waltham, MA, USA) with an Aminex 87H column (eluent: 0.01 N sulfuric acid, oven temperature: 40 °C, flow rate: 0.5 mL/min). Peaks were identified by comparing the retention time of each peak. The concentration of peaks was identified by comparing the standard calibration curve of each chemical. The furfural yield (Equation (1)) and pentose conversion (Equation (2)) were calculated as follows:

$$\text{Furfural yield (\%)} = \text{furfural after reaction (mol)}/\text{pentose before reaction (mol)} \times 100 \quad (1)$$

$$\text{Pentose conversion (\%)} = (\text{pentose before} - \text{after reaction (mol)})/\text{pentose before reaction (mol)} \times 100 \quad (2)$$

## 3. Results and Discussion

### 3.1. Pentose Production during Dilute Acid Pretreatment

*Quercus mongolica* is chemically composed of 46.67% of glucose, 19.14% of xylose, mannose and galactose (XMG), 0.77% of arabinose, 22.56% of acid insoluble lignin (AIL), 3.19% of acid soluble lignin (ASL), 2.06% of extractives, and 0.05% of ash. It is known that the glycosidic bond between hemicellulose-cellulose is cleaved, and the separated hemicellulose is converted to pentose-like XMG dilute acid pretreatment [22]. As shown in Table 1, the main component in dilute acid hydrolysate was XMG, and the main pentose was xylose [21]. Glucose and arabinose derived from arabinoxylan or glucuronoxylan were also detected; however, they were present in very small amounts compared with pentose. In sugar derivatives, the main product was acetic acid derived from the O-acetyl group in hemicellulose. Although furanic compounds such as 5-hydroxymethylfurfural (5-HMF) and furfural were produced by the acidic dehydration of released sugar, there was only a small amount of these compounds owing to the low severity of the dilute acid pretreatment. This pretreatment had not boosted the dehydration of sugar to the furanic [23].

### 3.2. RSM for Furfural Production from Xylose Standard Solution with Extracting Solvent

Furfural was produced from the xylose standard by acid-catalyzed dehydration, as listed in Table 2. In run #2 with a reaction temperature of 180 °C, over a 60 min duration using a xylose concentration of 10 g/L, the maximum furfural yield was 69.87% with 96.02% xylose conversion. Although the maximum xylose conversion was 99.32% at run #12, the furfural yield under these conditions was 68.12%; this is lower than the yield in run #2. These results indicate that the more severe experimental conditions of run #12 as a result of a longer reaction time induced the degradation of the furfural that had been produced. The lower furfural yield due to furfural degradation under severe experimental conditions was also observed in runs #4, 8, and 10; these runs were also characterized by long reaction times at certain temperatures. It is inferred that these reaction conditions caused furfural degradation despite the presence of the extraction solvent to prevent furfural degradation. However, a previous study has reported that the amount of furfural degradation due to severe experimental conditions when using an extracting solvent is marginal compared to the amount of furfural degradation in conditions with no extracting solvent [24]. The condensation of furfural was suppressed through the addition of the THF; as such, there were no insoluble precipitates detected from furfural condensation in all experimental conditions.

To evaluate the effect of each variable on furfural yield, regression analysis was undertaken using a $2^3$ factorial design matrix with corresponding furfural yield (%). The following quadratic equation was generated (Equation (3)), based on the outcomes of the regression analysis:

$$\text{Furfural yield (\%)} = -1122.2746 + 11.8539\,X_1 + 2.4079\,X_2 + 1.0355\,X_3 - 0.0091\,X_1X_2 - 0.0082\,X_1X_3 + 0.0015\,X_2X_3 - 0.0302\,X_1{}^2 - 0.0032\,X_2{}^2 + 0.0024\,X_3{}^2 \tag{3}$$

In the equation, $X_1$, $X_2$, and $X_3$ represent the actual reaction temperature, reaction time, and xylose concentration, respectively. The model had a high regression coefficient ($R^2 = 0.95$), indicating 95% variability in the response, while the *p*-value was extremely low (0.001), indicating that this regression model was significant. The coefficient of variation (CV) was 18.74%, which indicates the high precision and reliability of the experiments [25].

A three-dimensional (3D) plot and detailed contour of the RSM for furfural yield was established using Equation (3), by varying the three variables within the experimental range (Figure 1). As shown in Figure 1a, the furfural yield increased with reaction temperature and time to approximately 185 °C and 180 min, respectively. Once the temperature and time exceeded these points, there was a decrease in furfural yield due to its degradation under severe experimental conditions [26].

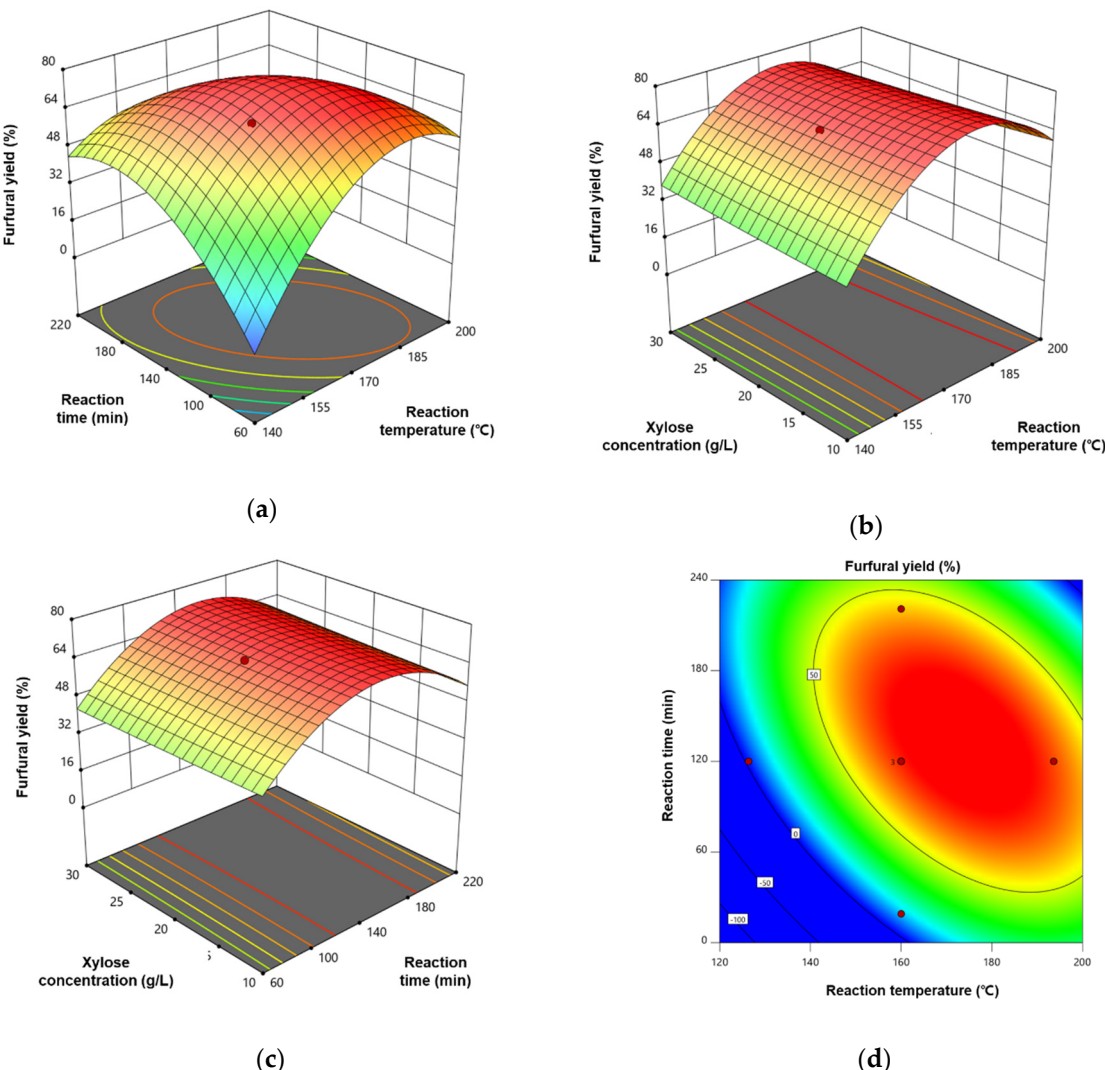

**Figure 1.** Three-dimensional plot and detailed contour of response surface methodology of furfural yield from xylose standard solution with extraction solvent (THF): (**a**) fixed xylose concentration (10 g/L); (**b**) fixed reaction time (120 min); (**c**) fixed reaction temperature (160 °C); (**d**) contour of furfural yield depending on reaction temperature and time.

This phenomenon was clearly observed in Figure 1b,c. Figure 1b depicts the furfural yield based on the reaction temperature and xylose concentration with a fixed reaction time of 120 min. The furfural yield increased with temperature until approximately 165 °C, and then remained stable at temperatures of 165–185 °C, which was where the maximum yield was observed. Then, yield decreased when the temperature increased above 185 °C regardless of the xylose concentration. These results are similar to the findings from previous research where furfural production from biomass occurred using sulfuric acid as a catalyst [27,28]. Figure 1c depicts the relationship between the furfural yield, the reaction time, and xylose concentration with a fixed reaction temperature of 160 °C. The furfural yield increased with reaction time up to 180 min. Beyond this time, yield decreased due to furfural decomposition [29] and self-condensation [30]. The xylose concentration was considered a less influential variable when compared to reaction temperature and time, as shown in Figure 1b,c. This inference was also supported by the p-value of the variables (Table 3).

The sources related to xylose concentration ($X_3$, $X_1X_3$, $X_2X_3$, $X_3^2$) had high *p*-values, indicating that xylose concentration was not as significant a factor for furfural yield. This result differs from previous research, which has demonstrated that furfural yield generally has an inverse relationship with xylose concentration. Higher xylose concentrations are able to produce more furfural at once, leading to a higher collision possibility that causes

condensation of the furfural reaction [31]. It was assumed that the 10–30 g/L xylose concentration range used in this study was too narrow of a range to affect furfural yield. This was unlike a study by Yang [32] where they produced furfural at high xylose concentrations. Yang [32] varied the xylose concentration from 40 to 120 g/L and found that furfural yield had been relatively stable when the xylose concentration was from 40 to 70 g/L. Based on the xylose concentration range used in this study (10–30 g/L), the effect on furfural yield was negligible compared to reaction time and temperature. With a fixed xylose concentration there was greater clarity in terms of the optimal conditions to maximize furfural yield. Additionally, the reaction temperature and time were expanded compared to the experimental range (Figure 1d).

**Table 3.** Analysis of variance (ANOVA) for furfural yield in dehydration of xylose model compound and coefficients for quadratic equation.

| Source | Sum of Squares | DF | Mean Square | *F*-Value | *p*-Value | Coefficient |
|---|---|---|---|---|---|---|
| Model | 10,697.21 | 9 | 1188.58 | 14.13 | 0.0010 | 67.77 |
| $X_1$ | 4722.88 | 1 | 4722.88 | 56.14 | 0.0001 | 18.60 |
| $X_2$ | 2136.23 | 1 | 2136.23 | 25.39 | 0.0015 | 12.51 |
| $X_3$ | 0.00 | 1 | 0.00 | 0.00 | 0.9963 | 0.01 |
| $X_1X_2$ | 955.35 | 1 | 955.35 | 11.36 | 0.0119 | −10.93 |
| $X_1X_3$ | 21.51 | 1 | 21.51 | 0.26 | 0.6287 | −1.64 |
| $X_1X_3$ | 6.48 | 1 | 6.48 | 0.08 | 0.7894 | 0.90 |
| $X_1{}^2$ | 1646.23 | 1 | 1646.23 | 19.57 | 0.0031 | −12.08 |
| $X_2{}^2$ | 1513.50 | 1 | 1513.50 | 17.99 | 0.0038 | −11.59 |
| $X_3{}^2$ | 0.67 | 1 | 0.67 | 0.01 | 0.9314 | 0.24 |
| Residual | 588.94 | 7 | 84.13 | | | |
| Lack of fit | 588.89 | 5 | 117.78 | 4825.36 | 0.0002 | |
| Pure error | 0.05 | 2 | 0.02 | | | |
| Corrected total | 11,286.15 | 16 | | | | |

The optimal conditions to maximize furfural yield was calculated based on Equation (3). The maximum furfural yield in the predicted reaction conditions was 75.1%, where the reaction temperature was 170 °C, reaction time was 120 min, and xylose concentration was 10 g/L. To verify the model, actual furfural production was carried out using these predicted optimal conditions, rendering a furfural yield of 72.39%, similar to the predicted yield.

### 3.3. Effect of Organic Solvent for Furfural Production and Extraction

Three kinds of organic solvents, THF, toluene, and DMSO, were evaluated to understand the effects of organic solvents on furfural production and extraction. Toluene is considered an effective solvent for furfural extraction [33], whereby it does not require additional salt for phase separation because of its immiscibility with water. DMSO has been used to improve the selectivity of 5-HMF from glucose by increasing the fructofuranose isomer and stabilizing 5-HMF by hydrogen bonding [34]. Under a similar mechanism, it was anticipated that DMSO could also improve furfural yield from xylose. Reaction conditions were set to the optimal values predicted from analysis of variance (ANOVA); this was a reaction temperature of 170 °C, reaction time of 120 min, xylose concentration of 10 g/L of, and the use of a 4% of sulfuric acid solution.

Table 4 shows the furfural yield from the xylose standard solution depending on the organic solvent. THF had the highest furfural yield from the xylose standard, while DMSO had the lowest among the three solvents. It was assumed that DMSO may not effectively protect the generated furfural from acid or water as furfural has no hydroxyl group compared with 5-HMF. In addition, DMSO has a reduced interaction with furfural compared to the other organic solvents such as toluene or THF, as its polarity is higher than that of furfural. THF had a higher furfural yield than toluene, even though the polarity of toluene was close to furfural. To explain this phenomenon, the partition coefficient of furfural in THF/water and toluene/water was calculated by dividing furfural

concentration in the organic solvent by the concentration of furfural in aqueous water (Table 4) to compare the solubility of furfural in a two-liquid mixture (water and organic solvent). To calculate the partition coefficient of furfural to THF/water, NaCl was added to separate THF from the water phase (salting out). THF had a higher partition coefficient than toluene, indicating that the former was more effective in extracting furfural than toluene, and yielding higher amounts of furfural.

**Table 4.** Properties of organic solvent and furfural yield from xylose standard based on the type of organic solvent utilized.

| Solvent | THF | Toluene | DMSO |
|---|---|---|---|
| Spectroscopic polarity (Furfural: 0.426 [35]) | 0.6 [36] | 0.55 [37] | 1 [38] |
| Partition coefficient * | 9.05 | 5.82 | N/D ** |
| Furfural yield (%) | 72.39 ± 0.50 | 58.01 ± 0.00 | 38.28 ± 0.00 |

\* Partition coefficient = [Furfural]org/[Furfural]aq, ** N/D: Not detected.

### 3.4. Production of Furfural from Pentose in Dilute Acid Hydrolysate

The optimal reaction conditions as predicted using the ANOVA (i.e., 170 °C, 120 min, and 10 g/L of xylose) were adopted to maximize furfural production from pentose in dilute acid hydrolysate. The xylose concentration was adjusted by mixing the hydrolysate, which had already contained 4% (w/w) sulfuric acid, with 4% sulfuric acid solution. The total volume of the aqueous phase was adjusted to 10 mL, similar to the xylose standard solution, and 20 mL of THF was added to extract furfural from the aqueous phase.

Table 5 describes the pentose conversion and furfural yield from the xylose standard and dilute acid hydrolysate. The furfural yield and pentose conversion of hydrolysate were slightly lower, compared with the xylose standard. The presence of impurities such as hexoses, organic acids, and acid soluble lignin, may have impacted on the extraction efficiency of furfural from the hydrolysate [39].

**Table 5.** Pentose conversion and furfural yield from xylose standard and dilute acid hydrolysate.

| | Pentose Conversion (%) | Furfural Yield (%) |
|---|---|---|
| Xylose standard solution | 100 ± 0.00 | 72.39 ± 0.50 |
| Liquid hydrolysate | 94.69 ± 0.76 | 68.20 ± 0.20 |

The difference in furfural yield between the xylose standard and the hydrolysate was not as considerable as had been expected. This indicates that THF effectively prevents furfural loss by inhibiting the ring opening of furfural and condensation between furfural and acid soluble lignin to form insoluble precipitate [40].

To investigate the distribution of impurities in the organic and aqueous phases, NaCl was added to separate the hydrolysate into the organic and aqueous phases. Table 6 presents the change in the distribution of furfural and other chemicals in furfural that produced hydrolysate by phase separation. Most furfural produced was extracted in the organic phase with a partition coefficient of 8.43; this is slightly lower than that of the xylose standard solution (9.05). Sugars, such as glucose, XMG, and arabinose favor the aqueous phase owing to their hydroxyl group, while other chemicals such as furfural, organic acids, and ASL tend to be extracted by the organic phase of THF. It is known that THF is able to effectively dissolve lignin as it has a high affinity to phenolic compounds [41], thus, most ASL was extracted to THF. Approximately three-quarters of organic acids were extracted to the organic phase, and the THF was effective in extracting various organic acids; however, the detailed extraction mechanism of THF to organic acid continues to be unclear [42,43].

**Table 6.** Concentration of chemicals in furfural produced liquor prior to and after phase separation (organic phase, aqueous phase) through the addition of NaCl.

| | Concentration (g/L) | | | | | | |
|---|---|---|---|---|---|---|---|
| | Furfural | Glucose | XMG | Arabinose | Formic Acid | Acetic Acid | Acid Soluble Lignin |
| Before separation | 3.08 ± 0.11 | 0.09 ± 0.02 | 0.22 ± 0.03 | 0.05 ± 0.02 | 0.49 ± 0.02 | 1.22 ± 0.03 | 0.84 ± 0.02 |
| Organic phase | 4.84 ± 0.10 | 0.00 ± 0.00 | 0.09 ± 0.00 | 0.00 ± 0.00 | 0.63 ± 0.02 | 1.61 ± 0.04 | 1.24 ± 0.00 |
| Aqueous phase | 0.58 ± 0.05 | 0.25 ± 0.05 | 0.29 ± 0.05 | 0.05 ± 0.00 | 0.21 ± 0.00 | 0.55 ± 0.03 | 0.21 ± 0.00 |

THF, the separated organic phase, was removed, and 20 mL of fresh THF was added to 10 mL of furfural extracted aqueous phase to enhance the extraction rate of furfural to the organic phase. After each additional extraction, the amount of extracted furfural and other compounds were analyzed; the extraction rate was calculated as (Equation (4)):

$$\text{Extraction rate (\%)} = \text{Amount of products extracted in THF phase (g)}/\text{Amount of products existed in furfural produced liquor before phase separation (g)} \tag{4}$$

Figure 2 shows the increase in the furfural extraction rate to THF with the number of extractions. Following the second additional extraction, the extraction rate increased from 86.03% to 94.63%. However, impurities such as organic acids and ASL had also been further extracted as the number of extractions increased. In particular, organic acids such as formic acid and acetic acid were completely extracted to THF following the second extraction. This means that additional impurity separation is required to acquire furfural with higher purity. It was reported that organic acids and ASL may be separated from THF by the ion exchange resin [44,45], and absorbents such as activated carbon, respectively [46]. However, lignin separation by an absorbent must occur prior to furfural production, as the absorbent absorbs furfural and ASL through a π-π interaction [47].

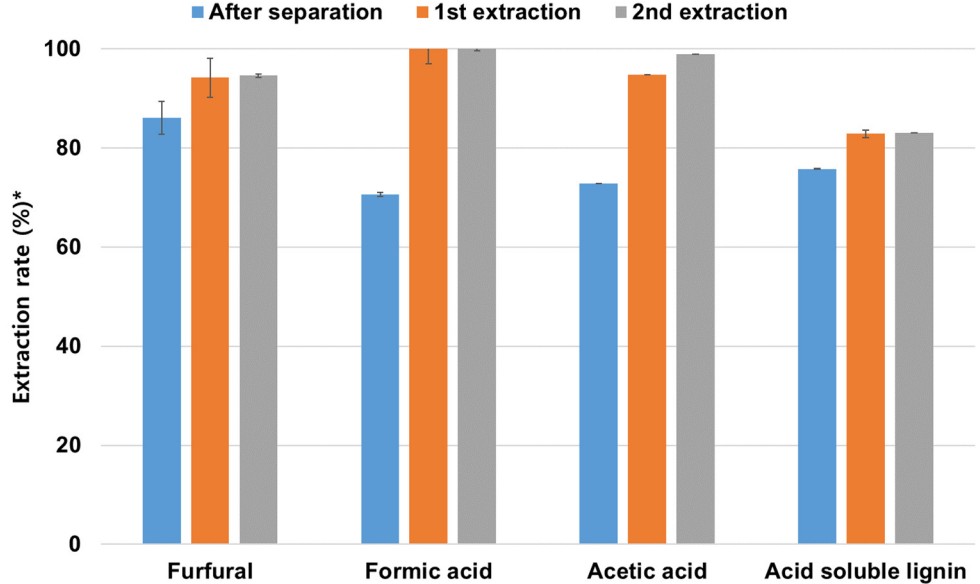

**Figure 2.** Extraction rate of products in furfural generated hydrolysate by extracting solvent (THF).

## 4. Conclusions

This study aimed to optimize furfural production from pentose in the dilute acid hydrolysate of *Quercus mongolica*. The main component of the acid hydrolysate was XMG, which was dominated by xylose. The optimization of furfural production was conducted using RSM with a xylose standard solution and an extracting solvent THF, to enhance furfural yield. The optimal conditions included a reaction temperature of 170 °C at a reaction time of 120 min with a xylose concentration of 10 g/L; the predicted furfural yield

was 75.1% under these conditions. An experimental furfural yield of 72.39% was obtained under the optimal experimental conditions, similar to the predicted yield. Extracted solvents such as THF, toluene, and DMSO were evaluated to understand the effect of the solvent on furfural yield. THF achieved the highest furfural yield, while DMSO had the lowest yield. Based on this result, furfural was produced from dilute acid hydrolysate under optimized reaction conditions using THF as the extracting solvent. A furfural yield of 68.20% was obtained based on pentose in the hydrolysate, similar to that of the xylose standard solution (72.39%). Following phase separation through the addition of NaCl, 86.03% of the furfural produced was in the organic phase. The THF, and two additional extractions using fresh THF enhanced the furfural extraction rate to 94.63%.

**Author Contributions:** Conceptualization, J.-H.K. and B.K.; methodology, S.-M.C. and J.-H.K.; formal analysis, J.-H.K. and J.-H.C.; investigation, J.-H.K.; resources, H.J. and S.M.L.; data curation, J.-H.K. and J.-H.C.; writing-original draft preparation, J.-H.K.; writing-review and editing, J.-H.K.; visualization, J.-H.K.; supervision, I.-G.C. and B.K.; project administration, B.K. All authors have read and agreed to the published version of the manuscript.

**Funding:** This study was carried out with the support of the R & D program for Forest Science Technology (Project No. 2020226C10-2022-AC01) provided by the Korea Forest Service (Korea Forestry Promotion Institute) and the Research Program (FP0900-2019-01) of the National Institute of Forest Science (NIFoS) (Seoul, Republic of Korea).

**Institutional Review Board Statement:** Not applicable.

**Informed Consent Statement:** Not applicable.

**Conflicts of Interest:** The authors declare no conflict of interest.

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
