# Peer review of "A Simultaneous Conversion and Extraction of Furfural from Pentose in Dilute Acid Hydrolysate of Quercus mongolica Using an Aqueous Biphasic System"

_applsci, doi:10.3390/app11010163_

Round 1
Reviewer 1 Report
The manuscript is well written easy to follow and has a good scientific quality. The conclusions are based on the scientific evidences.
I recommend its publication in this form.
Author Response
"Please see the attachment."

Reviewer 2 Report
This paper deals with one relevant issue: the selective production of aldehydes from xylose, preveting the furfural degradation or oligomerization. Although this is a hot point in the development of biomass-based refinery, there are so relevant weak points in the paper that this work is not prepared to being published. The main points to improve are the following ones:
1.- The English is so bad that it is almost impossible to understand the message of the work. There are many relevant mistakes related to tenses, use of prepositions, adjectives, articles, etc.
2.- The abstract must be rewritten. In addition to the English mistakes, the information is repeated and wrong organized.
3.- The introduction section is too long and general, without being focused on the objectives of this work. For example, which is the relevance of a supercritical water hydrolysis paragraph if this is not the technique used?
4.- Experimental section must include the procedures to obtain the chemical compositions, but not the results that must be part of the results and discussion section.
5.- There is not any relevance of all the decimal places shown in Table 3, and this type of study requires reproducibility tests to ensure the accuracy of model proposed.
6.- Carbon balance closures must be included in the data since there is not possible to distinguish if the use of a biphasic solvent prevents these undesired reactions. These data are required considering the complete miscibility of THF in water before adding the NaCl. This implies that reaction is taking place in water medium, so the authors must reinforce the advantage of using this methodology.
7.- The model proposed must be analyzed out of the range proposed in order to indicate if this is an universal fitting or if this equation is just a mathematical adjustement to some experimental data.
8.- I do not see any relevance of one hydrolysis model that does not consider the acid concentration as a key variable. THis is not a model but and adjustment of the experimental data. Including so many number of terms, a good adjustment is quite easy to obtain.
9.- The defense of this model would require testing other catalysts to corroborate if the adjustment is correct.
Author Response
"Please see the attachment."

Reviewer 3 Report
I think this is a very good work and should be published after following minor changes:
“size of raw material was reduced less than 0.5 mm by grinding and sieving” should change to: size of raw material was reduced to less than 0.5 mm by grinding and sieving. But the details should be described in a way that the work can be reproduced.
The following is not clear: “with 7 of solid to liquid ratio in Erlenmeyer flask”. Please describe well.
“Quercus mongolica at previous research.” Should change to: Quercus mongolica as described in the previous research. The authors should cite the previous papers that described this in detail.
Shouldn’t Table 1 move to results and discussion section? Also what are the uncertainties in the concentrations in this table?
Industrial application should be described in detail and should be put in the context in the following: “Considering industrial application, pentose released during dilute acid pretreatment, was…”
There should be a discussion about uncertainties of the results presented in all tables and ideally uncertainties should be added to them when possible.
Author Response
"Please see the attachment."

Reviewer 4 Report
This manuscript presents a thorough study of the conditions to recover furfural from Quercus mongolica. It is a good foundational study, but would benefit from a more clear statement as to how this fits in the larger schema. What are the next steps?
The abstract starts very weak. A purpose? The purpose would be stronger. When you state ‘a purpose’ it seems like you are then going to list the other purposes of the study.
The introduction focuses on lignocellulosic biomass and the issues associated with fractionation of the native compounds and removal of some important compounds. However, the present study is focused on furfural production from mongolian oak, which is not generally considered to be a major crop in the lignocellulosic biomass realm. Can you explain why you chose mongolian oak and potentially how the finding can be transferred to a more traditional lignocellulosic biomass source?
Table 2.2: Why were 126.36 and 193.64 °C chosen? These are so oddly precise and unlike the other temperatures in the table.
Table 3: Watch out for how many sigfigs you are presenting.
Author Response
"Please see the attachment."

Round 2
Reviewer 2 Report
Authors have improved the quality of this work based on the previous comments. However, there are still some relevant points that are not clear enough or must be completed before this work can be considered for publication.
1.- How many repetitions have been carried out? I mean for both, characterization and activity anayses. Results cannot be so accurate if they correspond to only one sample. Please indicate the standard deviation in all the results of table 1 and 2.
2.- Experiments at 126.36 and 193.64 °C have not any sense. Using mathematical methods to analyse and design experiments is a good idea, but the results must be analysed considering the real application. These experiments have nothing to do with an "applied" thing.
3.- Why the third column on table 5?
4.- I insist that the adjustments must be done according to physical reasons, and not only mathematical fits. I do not see the relevance of this mathematical adjustment and I consider that the good fit is only due to the high amount of parameters of this quadratic equation.
